# Therapeutic Relevance of Inducing Autophagy in β-Thalassemia

**DOI:** 10.3390/cells13110918

**Published:** 2024-05-25

**Authors:** Roberto Gambari, Alessia Finotti

**Affiliations:** Center “Chiara Gemmo and Elio Zago” for the Research on Thalassemia, Department of Life Sciences and Biotechnology, University of Ferrara, 44121 Ferrara, Italy; gam@unife.it

**Keywords:** autophagy, β-thalassemia, Ulk1, α-globin, ineffective erythropoiesis

## Abstract

The β-thalassemias are inherited genetic disorders affecting the hematopoietic system. In β-thalassemias, more than 350 mutations of the adult *β-globin* gene cause the low or absent production of adult hemoglobin (HbA). A clinical parameter affecting the physiology of erythroid cells is the excess of free α-globin. Possible experimental strategies for a reduction in excess free α-globin chains in β-thalassemia are CRISPR-Cas9-based genome editing of the *β-globin* gene, forcing “de novo” HbA production and fetal hemoglobin (HbF) induction. In addition, a reduction in excess free α-globin chains in β-thalassemia can be achieved by induction of the autophagic process. This process is regulated by the Unc-51-like kinase 1 (*Ulk1*) gene. The interplay with the PI3K/Akt/TOR pathway, with the activity of the α-globin stabilizing protein (AHSP) and the involvement of microRNAs in autophagy and *Ulk1* gene expression, is presented and discussed in the context of identifying novel biomarkers and potential therapeutic targets for β-thalassemia.

## 1. Introduction

The β-thalassemias are inherited genetic disorders affecting the hematopoietic system [1,2,3], characterized by more than 350 known mutations of the adult β-globin gene [4]. In addition to that, a clinical parameter affecting the physiology of erythroid cells is the excess of free α-globin [5,6,7,8,9,10]. Figure 1 summarizes the key biochemical/molecular alterations in β-thalassemia. In this pathology, the most important feature is ineffective erythropoiesis associated with a low or absent production of β-globin, leading to a low or absent production of HbA (α_2_β_2_) [1,2,3]. 

The low/absent production of β-globin is the leading cause of an accumulation of free α-globin chains in the cytoplasm of erythroid cells. This is the most important factor contributing to ineffective erythropoiesis [11,12]. In fact, the free α-globin chains tend to aggregate each other and precipitate, causing cytotoxic effects [5,13], oxidative stress [14] and low survival of the erythroid cells [15] associated with high levels of hemolysis [7,16].

From the analytical point of view, these alterations are easily appreciable using HPLC [17,18,19,20]. In this case, particularly informative is the comparison of chromatograms using lysates from erythroid cells derived from β^0^-thalassemia patients (who produce only trace amounts of HbA) and β^+^-thalassemia patients (who produce low but clearly detectable amounts of HbA). In both cases, the presence of α-globin aggregates is detectable. Figure 2 shows representative chromatograms of erythroid cells of a β^0^-thalassemia patient (Figure 2A) and a β^+^-thalassemia patient (Figure 2B).

In comparison with the HPLC profile of the blood from a healthy subject and considering HbA2 as a hemoglobin reference, absent (Figure 2A) and lower (Figure 2B) production of HbA is clearly evident, together with the presence of the “α-globin-peak”, located near the HbF peak and constituted by α-globin aggregates [21]. The insoluble fraction, composed of precipitated α-globin, can be studied by a different procedure. In this case, membranes are isolated and carefully washed to remove the soluble fraction. In the final step of the procedure, the proteins (that also contain the precipitated α-globin chains) are suspended in an SDS-containing buffer, separated by polyacrylamide gel electrophoresis and then analyzed by Western blotting to quantify the α-globins [21,22,23].

In this short review, we will focus on the interplay between the excess of the α-globin chain, ineffective erythropoiesis and autophagy in β-thalassemia. The interactions between the PI3K/Akt/TOR pathway, the activity of the α-hemoglobin stabilizing protein (AHSP) and the involvement of microRNAs in autophagy and *Ulk1* gene expression will also be presented and discussed in the context of identifying novel biomarkers and potential therapeutic targets for β-thalassemia.

## 2. The Clinical Impact of the Imbalanced Expression of *β-Globin* Genes Versus *α-Globin* Genes in β-Thalassemia: A Major Role for Ineffective Erythropoiesis in β-Thalassemia

The importance of studying the extent of excess α-globin chains in β-thalassemia is related to the fact that this information is clinically relevant since the free α-globin molecules, as schematically reported in Figure 1, are prone to precipitate, causing toxicity to the erythroid cells, interference with cell maturation [24,25] and ineffective erythropoiesis [11,12]. This is sustained by a number of complementary observations that concurrently demonstrate that α-globin expression is strictly associated with the severity of β-thalassemia. The first observation is that the co-inheritance of α-thalassemia traits significantly reduces the severity of β-thalassemia, as recently reviewed [26]. For instance, Gringras et al. showed that among the major determinants of β-thalassemia disease severity, the co-inheritance of α-thalassemia traits should be considered; in fact, the presence of an α-thalassemia deletion is associated with a significantly reduced initial disease severity [27]. The interaction of α-thalassemia with heterozygous β-thalassemia was extensively discussed by Kanavakis et al. [28], and the impact of α-globin expression on β-thalassemia was an object of several studies and reviews [27,28,29,30,31]; interestingly, silencing of α-globin can synergize with the induction of γ-globin in experimental protocols of interest as a potential therapy for β-thalassemia [32].

The clinical impact of the imbalanced expression of α-globin genes versus *β-globin* genes in β-thalassemia was further demonstrated using animal model systems. In a very informative study, β-globin knockout (KO) mice (beta^+/−^), characterized by a severe form of anemia, were used. In order to study the effects of the co-inheritance of α- and β-thalassemia, these mice were mated with heterozygous α-globin KO mice (alpha^++/−−^) [33]. The results obtained analyzing the phenotype of the alpha-KO/beta-KO mice were compatible with the following conclusions: (a) co-inheritance of α- and β-thalassemia in mice improves the thalassemic phenotype, as it was reported in humans; (b) the heterozygous murine β-globin KO mouse model might be a very useful in vivo model system to test druggable approaches for the therapeutic knockdown of *α-globin* gene expression. This was further demonstrated by the same research team, that published a study indicating a siRNA-mediated reduction of α-globin associated with phenotypic improvements in this β-thalassemic mouse model system [33].

A final proof-of-concept sustaining that a decreased expression of *α-globin* genes is beneficial for β-thalassemia was obtained using CRISPR-Cas9 protocols for decreasing the production of α-globin. For instance, CRISPR/Cas9 genome editing was employed by Mettananda et al. [34] to mimic a natural mutation causing α-thalassemia in association with a deletion of the *MCS-R2* α-globin enhancer. This protocol caused a dramatic reduction in the expression of *α-globin* genes [34]. Furthermore, in the report by Pavani et al. [35], the production of α-globin was downregulated by CRISPR/Cas9 editing, leading to *HBA2* gene deletion in hematopoietic cells from β-thalassemia patients [35]. This generated an α-thalassemia trait, correcting the pathological phenotype of β-thalassemia cells associated with an α/β-globin imbalance [35]. 

Figure 3 depicts additional approaches to reduce the imbalanced α-globin/β-like globin ratio in β-thalassemia. 

In this respect, Cosenza et al. [36] have demonstrated that the accumulation of corrected *β-globin* mRNA and a “de novo” production of β-globin and adult hemoglobin (HbA) were found after CRISPR-Cas9-based genome editing of the *β-globin* gene on erythroid cells from homozygous β^0^39-thalassemia patients. Interestingly, the CRISPR-Cas9-forced HbA production levels were associated with a significant reduction in excess free α-globin chains [36]. This strategy is indicated in Figure 3 as approach “A”. 

A reduction in excess free α-globin chains in β-thalassemia cells can also be obtained following *γ-globin* gene activation and fetal hemoglobin (HbF) induction (Figure 3, approaches “B” and “C”). In this case, increased *γ-globin* gene expression leads to an increased production of γ-globin. This contributes to decreasing free α-globin by the formation of α_2_γ_2_ tetramers (HbF). 

Evidently, γ-globin gene activation and fetal hemoglobin (HbF) induction can be obtained either by CRISPR-Cas9 gene editing, targeting γ-globin gene repressors or the binding sequences presenting the *γ-globin* gene promoter (Figure 3, approach “B”) [37,38,39], or by exposure of β-thalassemic erythroid cells to a variety of HbF inducers, (Figure 3C) as reviewed elsewhere [40,41,42] (Figure 3, approach “C”). Finally, a decrease in the excess free α-globin chains can be achieved in erythroid cells by the activation (or potentiation) of autophagy (as outlined in Figure 3, approach “D”), and is reported in several studies [43,44,45,46,47,48]. 

## 3. Reduction in Excess Free α-Globin in β-Thalassemia: The Role of Autophagy

In β-thalassemia, several pathways operate to prevent and/or counteract the deleterious event of excess free α-globin accumulation. First of all, it is clearly established that protein quality control (PQC) pathways, such as ubiquitin-mediated proteolysis, are activated in β-thalassemia [43,49]. In fact, free α-globin is stabilized by the AHSP (α-globin stabilizing protein); this facilitates solubilization and refolding [50,51]. The eventually misfolded α-globin undergoes poly-ubiquitinization and degradation in β-thalassemic erythroid cells (Figure 4, pathway A). This should, therefore, be considered an α-globin-specific proteolysis, increased in β-thalassemic erythroblasts relative to normal patient samples [52,53].

Interestingly, Khandros et al., using the heterozygous thalassemic (Th3/+) mouse model system, found an accumulation of insoluble α-globin chains in erythroid precursors that further accumulates in proteasome-inhibited β-thalassemic reticulocytes [43]. In addition, as shown in Figure 4, a key biological process leading to a reduction in excess free α-globin in erythroid cells is the autophagy process, which is dependent on the Unc-51-like autophagy-activating kinase 1 (Ulk1) [45,47,48].

### 3.1. Introductory Remarks on Autophagy

Autophagy is a highly regulated process that delivers proteins or organelles to lysosomes for degradation [54,55,56,57]. Autophagy pathways are essential for eukaryotic tissue development, cellular homeostasis and protection against metabolic and proteotoxic stresses [57,58,59,60,61]; in addition, autophagy is involved in several human diseases [61,62,63,64,65]. Figure 4 summarizes the key steps of the autophagy process, which can be tentatively sub-divided into initiation of the autophagosome (steps “a” and “b” of Figure 4B), vesicle nucleation, elongation and fusion (steps “c” and “d” of Figure 4B), and, finally, degradation (step “e”) [66].

Excellent reviews describing the biochemical/molecular pathways involved in autophagy are available [66,67,68,69,70]. As outlined in Figure 4, the initiation of autophagy is controlled by multi-protein complexes, the most important being the Unc1-like kinase 1 (Ulk1) complex. This ULK1 complex is constituted by Ulk1, ATG13, FIP200 (the FAK-family interacting protein of 200 kDa) and ATG101. The ULK1 complex is largely regulated by two sensor molecules, mTOR (mammalian target of rapamycin) and AMPK (AMP-activated protein kinase) [66,71,72]. The mTOR complex 1 (mTORC1), constituted by mTOR and raptor, in the presence of nutrients, inhibits autophagy following interaction with the ULK1 complex and phosphorylation of Ulk1 and ATG13 [66]. In Figure 4, other key players of autophagosome initiation and autophagosome maturation are indicated; among them, the rubicon (RUBCN) protein is of some interest, as it is a repressor of autophagosome maturation and autophagosome fusion with lysosomes [73,74,75]. The repressor activity of RUBCN is mediated by a direct binding of RUBCN to Beclin 1 complex and Rab7. In erythroid cells present in pathological conditions, among the stimuli known to activate the autophagy initiation are (a) the excess of free α-globin chains (as depicted in Figure 4), (b) the generation of ROS due to free iron, and (c) iron accumulation [66].

### 3.2. Autophagy in Human Diseases

In cancer, autophagy has been suggested to behave as a tumor suppressor as well a tumor promotion pathway [66,76,77,78,79]. Therefore, both inhibitors and activators of autophagy can be of interest in anti-cancer therapy. For instance, chloroquine (CQ) and its derivate hydroxychloroquine (HCQ) are known to inhibit autophagosome fusion with lysosome, inhibit tumor growth, both in vitro and in vivo, and have been approved by the FDA as anti-cancer agents [66]. On the other hand, considering the dual roles of autophagy in cancer, some autophagy activators are also suggested in cancer therapy, such as the mTOR inhibitor rapamycin [80] and the AMPK activator palbociclib [81]. In neurodegenerative diseases (such as Parkinson’s disease, Alzheimer’s disease, and Huntington’s disease), the role of the abnormal accumulation of certain neuroproteins is known [82,83]. In this respect, autophagy is critical for the degradation of protein aggregates in these diseases [84,85,86]. Accordingly, autophagy has a close interplay with neurodegeneration, as it can degrade toxic protein aggregates. Therefore, modulation of autophagy might be a possible therapeutic strategy for neurodegenerative diseases. A further example of autophagy in human diseases concerns infectious diseases. In this context, autophagy functions as a mechanism to fight against infectious agents, including bacteria (such as *Salmonella Typhimurium*) and viruses [66,87]. In the case of viral infections, autophagy can act both as an anti-viral or pro-viral mechanism [66]. Other examples of diseases associated with autophagy are several metabolic diseases, Crohn’s disease, and Lysosomal storage disorders (LSDs) [88].

### 3.3. Autophagy and Ineffective Erythropoiesis in β-Thalassemia

Autophagy participates in normal erythropoiesis, facilitating the removal of mitochondria, ribosomes and other organelles during terminal phases of differentiation of erythroid cells. The modulators of autophagy that regulate erythroid differentiation have been reviewed by Zhang et al. [89] and Grosso et al. [90]. In agreement, Mortensen et al. reported data showing that the autophagy protein Atg7 is essential for hematopoietic stem cell maintenance [91]. The same research group further supported this conclusion in a study based on mice lacking the *Atg7* gene. They found that these Atg7^−/−^ mice develop severe anemia, suggesting that accumulate of damaged mitochondria with altered membrane potential was responsible for the high frequency of erythrocyte cell death found in these mice [92]. The conclusion was that a loss of autophagy in erythroid cells leads to the defective removal of mitochondria and severe anemia when in vivo [92]. In this context, as already pointed out in Figure 1, improving ineffective erythropoiesis (IE) is mandatory in β-thalassemia, considering the clinical manifestation associated with IE in β-thalassemia, such as splenomegaly, as well as skeletal deformities due to extramedullary hematopoiesis, osteopenia, pulmonary hypertension, thrombosis, renal diseases, vascular diseases, endocrinopathies, liver diseases and cardiac diseases [93]. It is, therefore, not surprising that mitophagy was found to increase during erythroid differentiation in β-thalassemia [45,46,47,94]. Interestingly, increased autophagy leads to decreased apoptosis during β-thalassemic mouse and patient erythropoiesis [95], suggesting that a lack of autophagy potential might be associated with high-apoptosis of erythroid cells, a clear hallmark of ineffective erythropoiesis.

Finally, in the case of transfusion-dependent β-thalassemia, the role of iron in ineffective erythropoiesis and autophagy should be considered [96]. In this respect, the intracellular level of iron needs to be tightly balanced because an excess of iron (such as is occurring in β-thalassemia) can have damaging and heavily toxic effects caused by the generation of iron-catalyzed reactive oxygen species (ROS) [97]. In fact, iron can participate in the Fenton and Haber–Weiss reactions, leading to ROS generation and, thus, oxidative stress [97]. Autophagy is known to play an important role in maintaining physiological iron balance in the cell through its role in the degradation of the iron-storage protein, ferritin [98]. Interestingly, autophagy deficiency exacerbates iron-induced ROS production and apoptotic cell death [99]. This very important topic should also be considered with respect to the possible impact of iron chelators on the autophagic process, considering the use of these agents for the treatment of iron overload diseases, such as β-thalassemia [100].

## 4. The Unc-51-like Kinase 1 (*Ulk1)* Gene Plays an Essential Role in Regulating Autophagy in β-Thalassemia: Experimental Evidence from In Vivo Studies Involving β-Thalassemic Mice and β-Thalassemia Patients

Several recent pieces of evidence sustain the key role of the Unc-51-like kinase 1 (*Ulk1*) gene in regulating autophagy in β-thalassemia. The key study in this research field has been published by Lechauve et al. [45] and was designed and performed to determine whether Ulk1 was able to mediate free α-globin degradation in β-thalassemia and what the effects of Ulk1 activity were on the β-thalassemia phenotype. To this aim, they introduced a null allele into β-thalassemic mice (strain *Hbb*^Th3/+^) and compared the phenotype of mice carrying the following *Hbb*/*Ulk1* genotypes: (A) *Hbb*^+/+^ *Ulk1*^+/+^; (B) *Hbb*^Th3/+^ *Ulk1*^+/+^; and (C) *Hbb*^Th3/+^ *Ulk1*^−/−^. The *Hbb*^+/+^ *Ulk1*^+/+^ genotype exhibited a healthy phenotype; the *Hbb*^Th3/+^ *Ulk1*^+/+^ exhibited a β-thalassemia-like phenotype in the context of normal expression of the *Ulk1* gene; and the *Hbb*^Th3/+^ *Ulk1*^−/−^ genotype exhibited a β-thalassemia-like phenotype in the context of suppressed expression of the *Ulk1* gene. A summary of the most relevant results obtained by Lechauve at al. [45] is reported in Figure 5.

The results obtained conclusively show that the knocking out of *Ulk1* exacerbated β-thalassemia [45]. In *Hbb*^Th3/+^ *Ulk1*^−/−^ mice, a 20% reduction in the RBC count and a twofold increase in the reticulocyte count were observed in comparison to *Hbb*^Th3/+^
*Ulk1*^+/+^ mice. A loss of Ulk1 reduced the half-life of circulating β-thalassemia RBCs and caused increased splenomegaly, bone marrow hyperplasia, and extramedullary erythropoiesis in the spleen and liver. In addition, a loss of the expression of Ulk1 was associated with the accumulation of insoluble α-globin, a reduction in the RBC life span, worsening of ineffective erythropoiesis and decreased survival of *Hbb*^Th3/+^ *Ulk1*^−/−^ mice compared with *Hbb*^Th3/+^
*Ulk1*^+/+^ and *Hbb*^+/+^ *Ulk1*^+/+^ mice (Figure 5) [45]. 

A second important conclusion of the study by Lechauve et al. is that rapamycin is a pharmacologic inducer of Ulk1-mediated apoptosis in Hbb^Th3/+^ mice as well on erythroid precursor cells (ErPCs) from β-thalassemia patients [45]. 

In this respect, Zurlo et al. published a study indicating that *Ulk1* gene expression is upregulated, together with the induction of autophagy, following “ex vivo” exposure of ErPCs from β-thalassemia patients to low rapamycin concentrations (100–200 nM) (Figure 6) [47]. More importantly, they determined an increase in *Ulk1* gene expression in the ErPCs isolated from β-thalassemia patients participating in the NCT03877809 clinical trial and treated with 0.5–2 mg/day sirolimus [47].

Their data support the concept that autophagy, Ulk1 increased expression, and α-globin chain reduction are activated in β-thalassemia patients who take a daily dosage of 2 mg rapamycin (sirolimus). The experimental evidence and technology-transfer activities sustaining sirolimus-based clinical trials have been recently reviewed [101]; information on the NCT03877809 clinical trial can be found in Gamberini et al. [102], while the first report on the results of the trial has been published by Zuccato et al. [103]. The main results of the trial, reported by Zuccato et al. [103], were that the content of *γ-globin* RNA and HbF are increased in ErPCs isolated from sirolimus-treated β-thalassemia patients. No major side effects and no alteration of the immunophenotype were found. The results by Zurlo et al. [47], depicted in Figure 6 (panels E and F), suggest that *Ulk1* gene expression is upregulated, and the excess of free α-globin decreases in the ErPCs isolated from sirolimus-treated β-thalassemia patients, supporting the conclusions originally presented by Lechauve et al. [45] regarding the fact that the autophagy-activating kinase Ulk1, inducible by rapamycin, mediates clearance of free α-globin in β-thalassemia mice, improving the phenotype (see also Figure 5).

## 5. MicroRNAs as Possible Targets for Induction of Autophagy in Erythroid Cells of β-Thalassemia Patients: A Working Hypothesis

MicroRNAs (miRNAs) are a class of non-protein-encoding RNA molecules that specifically bind to the 3′-untranslated region (3′-UTR) of target mRNAs, causing their degradation or protein translation inhibition to maintain optimal levels of the target protein [104,105,106,107]. There is a large consensus on the fact that miRNAs are involved in the regulation of a variety of cellular and molecular events, including cell proliferation, differentiation, metabolism and apoptosis. Several review articles on miRNAs’ biological functions in normal and pathological cells and tissues are available [108,109,110,111,112,113].

The roles of miRNAs during the different phases of autophagy have been presented and discussed in several reports [114,115,116,117,118]. MicroRNAs are involved in all the phases of autophagy. This has been extensively studied in cancer, cardiovascular disorders, diabetes, digestive and kidney diseases, infectious diseases, inflammation, osteoarthritis and neurological disorders [119]. For instance, miR-144, miR-125a, miR-761, miR-129-3p, miR-146a, miR-126, miR-494, miR-22, miR-145, and miR-99b are all miRNA that are involved in transcriptional silencing of the PI3K/Akt/TOR networks that are responsible for the downregulation of autophagy [119,120,121,122,123,124,125,126,127]. Beclin-1, ATG12, ATG5, ATG7, ATG14, ATG2B and ATG16L1 (all activators of autophagy) are transcriptionally silenced by miR-129-5p and miR-30a (Beclin-1) [114,128,129]; miR-23b (ATG12) [130]; miR-30a, miR-30c, miR-376b, miR-153-3p and miR-46a-5p (ATG5) [119,131,132,133]; miR-153-3p, miR-202-5p, miR-210, miR-375 and miR-17 (ATG7) [119,134,135,136]; miR-375 and miR-25-3p (ATG14) [137,138]; miR-375 (ATG2B) [119,139]; and miR-130a and miR-20a-5p (ATG16L1) [119,140].

With respect to the relevance for possible applications is in inducing Ulk1 and, consequently, autophagy, several microRNAs have been demonstrated to be involved in the post-transcriptional regulation of *Ulk1* mRNA [141,142,143,144,145,146,147,148,149]. For instance, Wu et al. reported that miR-20a and miR-106b negatively regulate autophagy that has been induced by leucine deprivation via the suppression of Ulk1 expression in C2C12 myoblasts [141]. A second example is the study by Zheng et al. [142], who were able to demonstrate that miR-26a-5p regulates cardiac fibroblasts’ collagen expression by targeting *Ulk1*. Furthermore, Chen et al. reported that the downregulation of Ulk1 by microRNA-372 inhibits the survival of human pancreatic adenocarcinoma cells [147]. Considering the key role of Ulk1 in an autophagy-mediated decrease of excess α-globin chains in β-thalassemia (Figure 4) and considering the important role of miR-144/451 in the regulation of gene expression during mouse erythropoiesis [150], Keith et al. demonstrated that the disruption of the bicistronic microRNA gene miR-144/451 alleviates β-thalassemia by reducing mTORC1 activity and stimulating Ulk1-mediated autophagy of free α-globin [48]. Interestingly, they provided a strong demonstration that miR-144/451 indirectly regulates Ulk1. In fact, the disruption of miR-144/451 increases Cab39/Strad/LKB1 activation of AMPK and increases the autophagy of α-globin via activation of the Ulk1 autophagy kinase, improving cell survival [48]. This very interesting finding was recently discussed by Babbs [151] and opens new perspectives in pharmaceutic control of free α-globin degradation and the counteraction of ineffective erythropoiesis.

## 6. Discussion

β-thalassemias are inherited genetic disorders affecting the hematopoietic system and are caused by a low or absent production of adult hemoglobin (HbA) [1,2,3]. A clinical parameter affecting the pathophysiology of erythroid cells is excess free α-globin. This is the most important factor contributing to ineffective erythropoiesis [11,12]. In fact, the free α-globin chains tend to aggregate each other and precipitate, causing cytotoxic effects, oxidative stress and low survival of the erythroid cells [15], which is associated with high levels of hemolysis [14]. Possible experimental strategies for a reduction in excess free α-globin chains in β-thalassemia are CRISPR-Cas9-based genome editing of the *β-globin* gene, forcing “de novo” HbA production levels and fetal hemoglobin (HbF) induction [36].

Table 1 summarizes the strategies used to reduce the content of α-globin, thereby ameliorating the β-thalassemia phenotype and the α/β-globin chain ratio.

In this review, we present studies supporting the involvement of autophagy in the reduction of excess α-globin production in β-thalassemia. This process is highly regulated by the Unc-51-like kinase 1 (*Ulk1*) gene, as demonstrated by the study published by Lechauve et al. [45]. The results obtained by this group conclusively show that the knocking out of Ulk1 in the β-thalassemia Th3/+ mouse model system exacerbates β-thalassemia [45]. The most important results of this study are summarized in Figure 5. Most of the studied hematological parameters that are a typical feature of the β-thalassemia Th3/+ mice (such as ineffective erythropoiesis, presence of insoluble α-globin, spleen size, half-life of circulation RBC and mean survival) are worsened when mice with the *Hbb*^Th3/+^ *Ulk1*^−/−^ genotype were compared to mice with the *Hbb*^Th3/+^
*Ulk1*^+/+^ genotypes [45]. A second conclusion of Lechauve’s study was that the mTOR inhibitor rapamycin was able to induce Ulk1 and autophagy, decreasing excess α-globin chains in the erythroid precursors isolated from β-thalassemia patients and treated “ex vivo” with rapamycin (sirolimus). In this respect, Zurlo et al. published a study that fully supports the study by Lechauve et al. [45], indicating that *Ulk1* gene expression is upregulated “in vivo” in the ErPCs isolated from β-thalassemia patients participating in the NCT03877809 clinical trial and treated with 0.5–2 mg/day sirolimus [47]. In conclusion, the induction of *Ulk1* gene expression by the mTOR inhibitor rapamycin (sirolimus) was demonstrated both “ex vivo” [45,47] and “in vivo” [47]. The understanding of the interplay of the PI3K/Akt/TOR pathway, expression of Ulk1 and autophagy (see Figure 4 for a brief scheme) is important not only from the theoretical point of view but also from the applied point of view in biomedicine.

Rapamycin (sirolimus) has been proposed as an in vivo fetal hemoglobin inducer for the treatment of β-thalassemia. Information on rapamycin, from its discovery and the first applied biomedical studies to the in vitro, ex vivo and in vivo experiments demonstrating its possible effect as HbF inducer, can be found in the review article by Gambari et al. [101]. Considering the rapamycin-mediated effects on HbF production, two clinical trials have been proposed for β-thalassemia using sirolimus as a therapeutic drug, NCT03877809 and NCT04247750 [102,103], with the objective of verifying the in vivo effects of treatment with low dosage of sirolimus, with respect to fetal hemoglobin production and expression of *γ-globin* genes in erythroid cells.

In this respect, the study by Lechauve et al. [45] and by Zurlo et al. [47] support the hypothesis that, in addition to the induction of fetal hemoglobin, sirolimus reduces the excess free α-globin chains by activating autophagy via *Ulk1* gene expression, both in vitro and in vivo. A reduction in the excess insoluble free α-globin chains and the expression of autophagy-related genes (including *Ulk1*) should be considered, in addition to γ-globin and HbF production, as endpoints in sirolimus-based clinical trials for β-thalassemias, such as the NCT03877809 and NCT04247750 clinical trials.

To study the interplay between Ulk1, autophagy and the control of excess α-globin production, novel experimental model systems might be of interest. In this respect, the experimental mouse system was carrying the *Hbb*^Th3/+^ *Ulk1*^−/−^ genotype, as described by Lechauve et al. [45]. With respect to the “in vitro” model systems, Zurlo et al. have recently obtained K562 cellular clones that are able to hyper-express α-globin protein [153]. In this work, they have demonstrated that the accumulation of toxic α-globin is able to induce apoptosis proportionally to its production/accumulation rate. Interestingly, they found that autophagy is spontaneously triggered by the accumulation of α-globin protein, and cellular clones that produce higher levels of the protein seem to have higher levels of p62 adaptor protein, probably in order to manage α-globin clearance. This and similar model systems might be of interest for further studying the regulation of the expression of the *Ulk1* gene, including the possible involvement of microRNAs in autophagy and *Ulk1* gene expression [153].

A further model system is represented by the ErPCs isolated from β-thalassemia patients. These cells have been applied for the identification and characterization of fetal hemoglobin inducers, can be bio-banked, and can be corrected either by gene therapy protocols or by gene editing. The preliminary observation published by Zurlo et al. demonstrates that basal levels of Ulk1 expression (analyzed by RT-qPCR) are highly variable (see Figure 5D). In addition, different responses to rapamycin (sirolimus) treatment “in vivo” were observed [47]. These ex vivo cultured ErPCs from β-thalassemia patients might, therefore, be considered for studying the effects of differential expression of Ulk1 on the other autophagy-associated biochemical and/or molecular parameters.

Some of the results obtained using the ErPC model system that was focused on the interplay between Ulk1 and autophagy are already available. Zurlo et al. reported that low concentrations of sirolimus (100 and 200 nM) are able to induce Ulk1 and reduce p62, strongly suggesting the activation of autophagy in the erythroid precursor cells (ErPCs) from β-thalassemia patients [47] This was confirmed using the ErPCs from sirolimus-treated β-thalassemia patients who participated in the NCT03877809 (Sirthalaclin) clinical trial. In these ErPCs, Ulk1 was found to be upregulated, and the excess of free α-globin chains was strongly reduced. In our opinion, the results on Ulk1 expression in the ErPCs isolated from sirolimus-treated patients deserve attention since the role of Ulk1 in the pathophysiology of β-thalassemia is a topic of active interest and translational applications.

In this context, no information is available in the literature about the possible co-activation of the AHSP-dependent and Ulk1-dependent pathways in sirolimus-treated erythroid cells from β-thalassemia patients. This might be considered for further studies in order to verify the very interesting possibilities of a co-regulation of transcription of *AHSP* and the *Ulk1* gene in erythroid cells. Studies of interest in this research area are those demonstrating that the transcription factor NF-E2 positively regulates *Ulk1* [154] and the *AHSP* [155] genes.

Finally, with respect to the possible therapeutic approaches targeting autophagy and ineffective erythropoiesis, some strategies have been presented in Table 1. In most of them, erythroid-lineage-specific autophagy-targeted therapy is a very challenging issue. Oligonucleotide-based therapy should take advantage of efficient (and possibly erythroid-selective) delivery systems. With respect to this issue, the choice of a good molecular target is, of course, important. In the case of targeting α-globin RNA, the selectivity of this intervention is expected to be high due to the tissue-specific expression of the γ-globin gene. On the contrary, many biochemical and molecular targets of autophagy are expressed in several tissues (for instance, the autophagy inhibitors mTOR, mTOR-associated raptor and autophagy inhibitor Rubicon (RUBCN) (see Figure 4)), supporting the concept that approaches modulating these targets might affect several tissues. In this context, a very interesting approach for controlling autophagy in erythroid cells is based on the modulation of GATA-1, which is considered a master regulator of erythropoiesis [156,157]. At the same time, Chromatin immunoprecipitation (ChIP) assays revealed GATA-1 occupancy at the regulatory sites of many autophagy genes [158]. The molecular approaches demonstrated that GATA-1 regulates autophagy [158]. Accordingly, studies focusing on the interactions between GATA-1 and other transcription factors and modulators might open new avenues for erythroid-specific approaches regulating autophagy.

## 7. Conclusions and Future Perspectives

In this review article, we presented and discussed several studies outlining the relationship between a reduction in excess α-globin in β-thalassemia and the activation of Ulk1-mediated autophagy. Future studies are needed in order to identify novel and potent inducers of autophagy in β-thalassemia. In this respect, rapamycin (sirolimus) appears of great interest for the following reasons: (a) it is a direct down-regulator of mTOR and, therefore, is able to upregulate Ulk1 expression and autophagy (Figure 4); (b) it is a potent HbF inducer, both in vitro and in vivo; (c) it is considered in two clinical trials in β-thalassemia (the NCT03877809 and NCT04247750 clinical trials); and (d) it upregulates Ulk1 in treated β-thalassemia patients.

A second future perspective is the development of a combined treatment using inducers of Ulk1-mediated autophagy and other approaches for therapy of β-thalassemia, including gene therapy and gene editing, to obtain the de novo production of HbA together with the highest possible reduction of excess free α-globin chains. A combination of HbF induction with the induction of Ulk1-mediated autophagy could be considered of interest.

A final comment concerns the interplay between AHSP activity and *Ulk1* gene expression. Both pathways are involved in a reduction of α-globin excess in erythroid cells (see Figure 4). It is, in fact, firmly established that the α-hemoglobin-stabilizing protein, AHSP, a chaperone highly expressed in erythroid cells, is involved in counteracting α-globin precipitation and related cytotoxicity [50,51].

## Figures and Tables

**Figure 1 cells-13-00918-f001:**
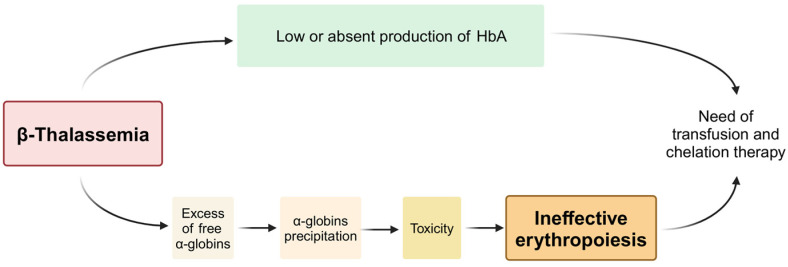
The interplay between production of HbA, excess of free α-globin chains and ineffective erythropoiesis in β-thalassemia. Created with BioRender.com.

**Figure 2 cells-13-00918-f002:**
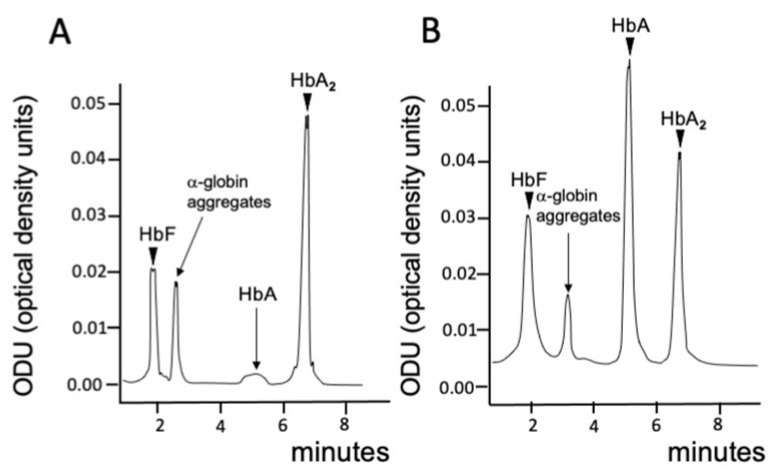
Representative HPLC profiles showing the presence of an “α-globin-peak” (“α-globin aggregates” close to the HbF peak) (**A**). HPLC of erythroid cells from a β^0^39/β^0^39 patient. (**B**). HPLC of erythroid cells from a β^0^39/β^+^-IVSI-110 patient. Modified from Cosenza et al. [17] (panel **A**) and Zuccato et al. [18] (panel **B**) with permission (copyright can be found at https://www.mdpi.com/2073-4425/13/10/1727 and https://www.mdpi.com/1420-3049/29/1/8, respectively; accessed on 25 March 2024) [18,20].

**Figure 3 cells-13-00918-f003:**
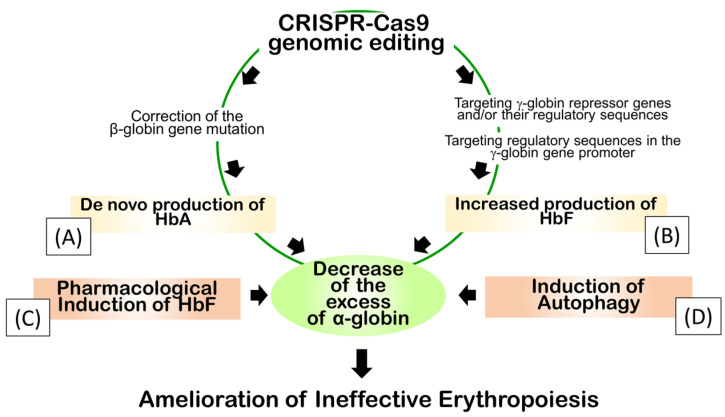
Possible experimental strategies for reduction in excess free α-globin chains in β-thalassemia. (**A**). CRISPR-Cas9-based genome editing of *β-globin* gene, forcing “de novo” HbA production levels. (**B**). Fetal hemoglobin (HbF) induction obtained by CRISPR-Cas9 gene editing. (**C**). Fetal hemoglobin (HbF) induction obtained by exposure of β-thalassemic erythroid cells to HbF inducers. (**D**). Induction of the autophagic process. Created with BioRender.com.

**Figure 4 cells-13-00918-f004:**
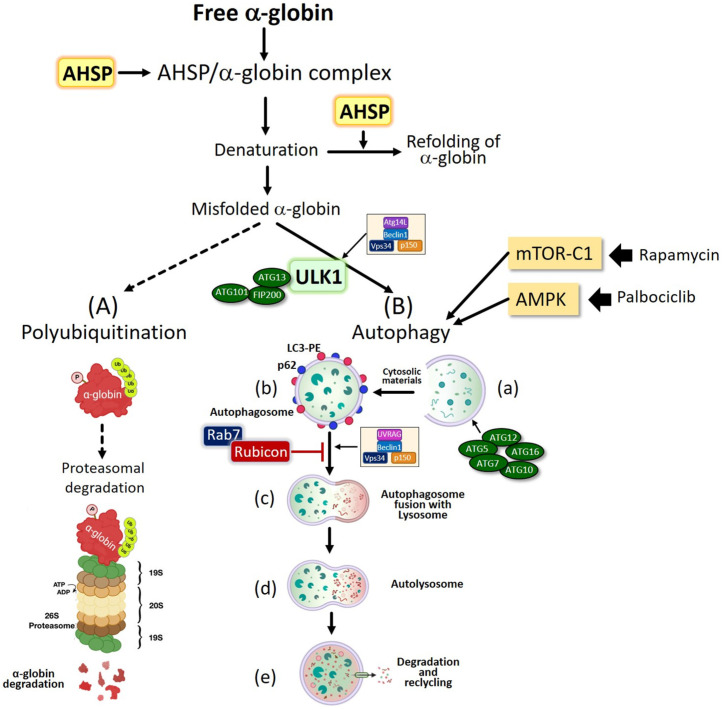
Pathways operating in erythroid cells to reduce the excess free α-globin in β-thalassemia. (**A**) Proteasome-mediated degradation of polyubiquitinated misfolded α-globin chains; (**B**) autophagy-mediated Ulk1-dependent pathway. The different phases of autophagy are indicated (**a**–**e**). Created with BioRender.com.

**Figure 5 cells-13-00918-f005:**
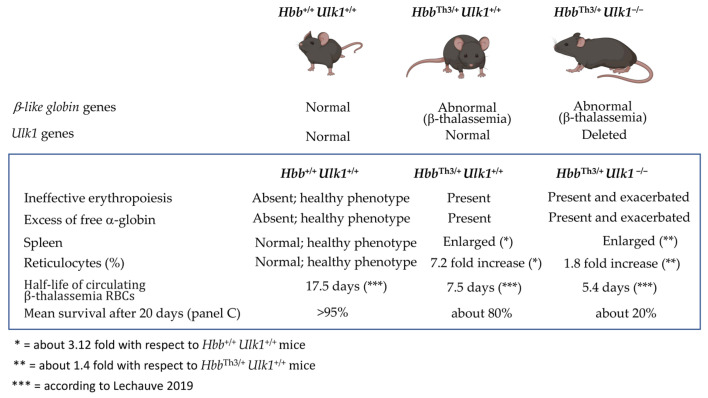
Summary of the key results of the study by Lechauve et al. [45] demonstrating that Ulk1 mediates clearance of free α-globin in β-thalassemia phenotypic differences in mice carrying the *Hbb*^+/+^
*Ulk1*^+/+^, *Hbb* ^Th3/+^
*Ulk1*^+/+^, and *Hbb*^Th3/+^ *Ulk1*^−/−^ genotypes are shown in the upper part of the panel. Created with BioRender.com.

**Figure 6 cells-13-00918-f006:**
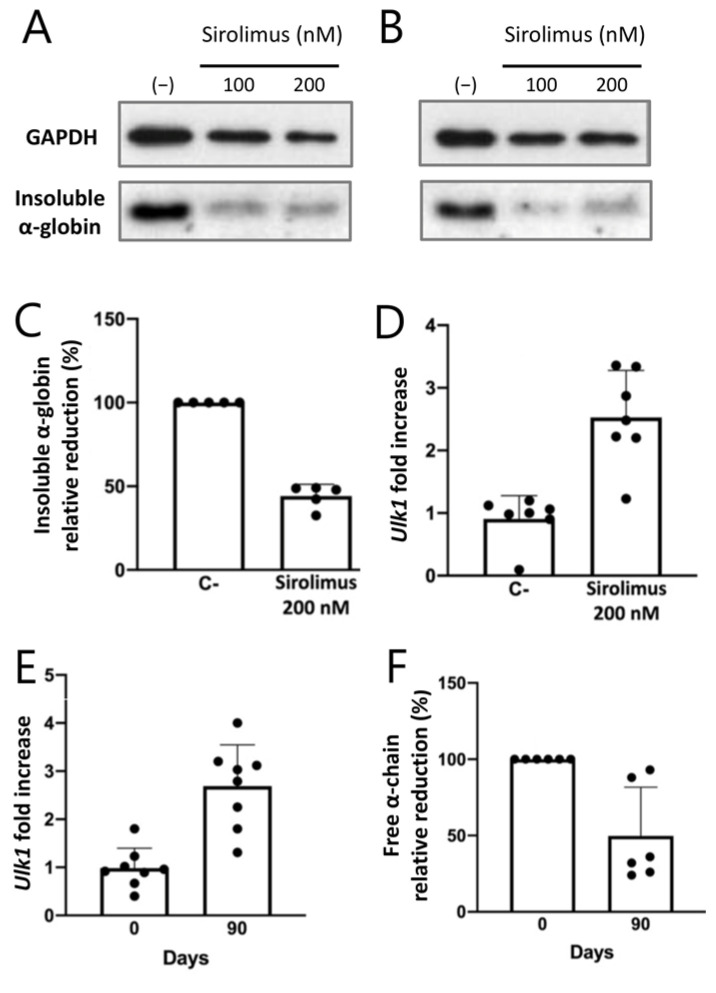
Key results of the study by Zurlo et al. demonstrating Ulk1 upregulation in β-thalassemia patients treated with rapamycin (sirolimus) [47]. (**A**). Rapamycin reduces insoluble α-globin in erythroid precursor cells treated with 100 and 200 nM sirolimus. This is further confirmed in panel (**B**), in which the results from 5 ErPC cultures are summarized. The decrease in insoluble α-globin is associated with an increased content of *Ulk1* mRNA (panel (**C**)). (**D**–**F**). Increased *Ulk1* gene expression (**D**,**E**) and a decrease in free α-globin chains (**F**) are present in ErPCs isolated from β-thalassemia patients treated with sirolimus and participating in the NCT03877809 clinical trial. Modified from Zurlo et al., with permission (copyright can be found at https://www.mdpi.com/1422-0067/24/20/15049, accessed on 3 March 2024) [47]. The uncropped version of the Western blots shown in panels (**A**,**B**) can be downloaded at https://www.mdpi.com/1422-0067/24/20/15049, accessed on 9 May 2024.

**Table 1 cells-13-00918-t001:** Examples of experimental approaches to reduce the excess α-globin production in β-thalassemia.

Approach and/or Drug	Key Results of the Reviewed Study	Authors	Comments
siRNA targeting	siRNA-mediated reduction of α-globin chain was obtained after transfection of in vitro β-thalassemia murine primary erythroid cells with siRNA targeting the *α-globin* mRNA.	Voon et al., 2008 [29]	This study indicates that siRNA-mediated reduction of α-globin has potential therapeutic potentials for β-thalassemia.
RNAi, antisense RNA	A combined RNAi and antisense RNA approach targeting *α-globin* mRNA has been used to restore balanced *α/β-globin* gene expression in β654-thalassemia mice.	Xie S.et al, 2007 [12]	As amelioration of hematologic parameters was observed in treated mice, the data presented in this study demonstrate the feasibility of this approach for β-thalassemia therapy by balancing α- and β-globin chains.
Treatment with antagomiRNAs	This study demonstrates that loss of miR-144/451 alleviates β-thalassemia by stimulating Ulk1-mediated autophagy of free α-globin.	Keith et al., 2023 [48]	The microRNA miR-144/451 should be considered a potential target of druggable anti-miRNA molecules, with the objective of reproducing the upregulation of Ulk1 functions, the induction of autophagy and improving the β-thalassemia phenotype of treated cells.
CRISPR/Cas9 genome editing	CRISP/Cas9-based editing of an *α-globin* enhancer in primary human hematopoietic stem cells was employed to reduce α-globin expression, showing effectiveness in xenograft assays in mice.	Mettananda et al., 2017 [34]	This study demonstrates the feasibility of editing an *α-globin* enhancer in primary human hematopoietic stem cells as a treatment for β-thalassemia.
CRISPR/Cas9 genome editing	Correction of β-thalassemia by CRISPR/Cas9 editing of the *α-globin* locus in human hematopoietic stem cells.	Pavani et al., 2021 [35]	Overall, we described an innovative CRISPR/Cas9 approach to improve α/β-globin imbalance in thalassemic HSPCs, paving the way for novel therapeutic strategies for β-thal.
CRISPR/Cas9 genome editing	CRISPR-Cas9-based genome editing of *β-globin* gene was employed on erythroid cells from homozygous β^0^39-thalassemia patients.	Cosenza et al., 2021 [36]	The CRISPR-Cas9-based strategy forced HbA production associated with a significant reduction in excess free α-globin chains.
Cas9/AAV6-mediated genome editing	The employed Cas9/AAV6-mediated genome editing strategy can replace the entire *HBA1* gene with a full-length *HBB* transgene in β-thalassemia-derived hematopoietic stem and progenitor cells.	Cromer et al., 2021 [152]	Gene replacement of *α-globin* with *β-globin* restores hemoglobin balance in β-thalassemia-derived hematopoietic stem and progenitor cells.
Treatment with vorinostat	This histone deacetylase inhibitor drug, vorinostat, in addition to its beneficial effects for patients with β-thalassaemia through induction of γ-globin, has the potential to simultaneously suppress α-globin expression.	Mettananda et al., 2019 [32]	A randomized clinical trial for evaluating the efficacy of Vorinostat to induce fetal hemoglobin in sickle cell disease has been proposed (NCT01000155).
Treatment with rapamycin	Rapamycin induces the autophagy-activating kinase Ulk1, which mediates clearance of free α-globin in β-thalassemia.	Lechauve et al., 2019 [45]	Ulk1 can be proposed as a functional target to mediate clearance of free α-globin in β-thalassemia.
Treatment with rapamycin	Decrease in α-globin and increase in the autophagy-activating kinase *Ulk1* mRNA in erythroid precursors from β-thalassemia patients treated with rapamycin.	Zurlo et al., 2023 [47]	Two clinical trials on β-thalassemia have been proposed, useful to further clarify this issue (NCT03877809 and NCT04247750).

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
