# Peer review of "Therapeutic Relevance of Inducing Autophagy in β-Thalassemia"

_cells, 2024, doi:10.3390/cells13110918_

Round 1

Reviewer 1 Report

Comments and Suggestions for Authors

The manuscript titled, “Therapeutic relevance of inducing autophagy in b-thalassemia” is interesting but it has some concerns as follows:

1.       The manuscript title talks about the relevance of autophagy induction, but in the entire manuscript the role of autophagy is dealt with in an extreme superficial manner. It seems the manuscript is revolving around a few handful autophagy based original research paper.

2.       Fig 3 should include the molecular players of autophagy steps.

3.       The iThenticate score is 34% indicating noticeable similarity in many parts of the manuscript.

4. Clinical data is not discussed in detailed manner.

Author Response

Reply to Reviewer #1.

First of all, we like to thank the reviewer for her(his) work.

General comments. The manuscript titled, “Therapeutic relevance of inducing autophagy in b-thalassemia” is interesting but it has some concerns.

Answer to the general comments. Thank you for your work, hoping to have fairly addressed your comments.

Point 1. The manuscript title talks about the relevance of autophagy induction, but in the entire manuscript the role of autophagy is dealt with in an extremely superficial manner. It seems the manuscript is revolving around a few handful autophagy-based original research paper.

Answer to point 1. Thank you for this comment. We agree on the fact that more information on the biochemical/molecular basis of autophagy would be important for the reader. On page 2, lines 58-63 we clarified the objectives of this short review, i.e. a focus on the interplay between the excess of a-globin chain, ineffective erythropoiesis, and autophagy in b-thalassemia. To give more information on autophagy, chapter 3.1 has been included (pages 4 and 5, lines 149-178. In addition, Figure 4 has been implemented (page 5). The new references 66-75 have been included to sustain the text.

Point 2. Fig 3 should include the molecular players of autophagy steps.

Answer to point 2. Figure 4 has been implemented (page 5), including molecular players of autophagy steps. Figure 4 has been described in lines 153-178.

Point 3. The iThenticate score is 34% indicating noticeable similarity in many parts of the manuscript.

Answer to point 3. We went through the similarity reports (thanks), and we tried to eliminate duplications. We understand that limiting similarity is important, and we will work to further decrease similarity if our work will be considered not sufficient. We have only considered similarities within the text. All the changes have been red-marked.

Point 4. Clinical data is not discussed in detailed manner.

Answer to point 4. The clinical data concerning rapamycin in beta-thalassemia have been discussed on page 8, lines 277-286, by including the sentence “The main results of the trial, reported by Zuccato et al. [103], were that the content of gamma-globin RNA and HbF are increased ….. mediates clearance of free α-globin in β-thalassemia mice, improving the phenotype (see also Figure 5)”. In addition, information of the clinical trials with rapamycin can be found in the following sentence “Information on rapamycin, from its discovery and the first applied biomedical studies, …… NCT03877809 and NCT04247750 [102,103], with the objective of verifying the effects in vivo of treatment with low dosage of sirolimus, with respect to fetal hemoglobin production and expression of γ-globin genes in erythroid cells” (page 11, lines 373-380).  

In conclusion, we thank the reviewers for the hard work and for the suggestions, that were judged to be very useful for improving the scientific quality of our review paper. We hope that this manuscript will be now considered acceptable for publication in the Special Issue “Exclusive Review Papers in Autophagy-Second Edition” (Cells).

Waiting for your comments, we like to thank you in advance for your help.

Alessia Finotti and Roberto Gambari,

Department of Life Sciences and Biotechnology

University of Ferrara

Ferrara, May 9, 2024

Reviewer 2 Report

Comments and Suggestions for Authors

This manuscript describes the new approach for beta-Thalassemia using “Autophagy system” in erythroid cells as biomarkers or therapeutic targets. The aim and scope are important and precious for contributing to sustainable health care of patients with beta-Thalassemia. Structure of main part of the manuscript is logical and scientific enough. Many important previous reports are cited appropriately. Writing is excellent, easily understandable for scientists in broad fields. This review would give us many relevant knowledges and encouragement to contribute to improve patient’s life.

Major points:

1.     Lines 181-238: Dysregulation of autophagy system in beta-Thalassemia is written only in abstract phrase. It is better to show data of up-regulation of autophagy in beta-Thalassemia accumulated in previous literatures.

2.     In general, there are few presentations for the molecular mechanism of autophagy. This is critical for understanding the cutting-edge of beta-Thalassemia treatment. We can see important progress in animal model and clinical trial in human in several parts of text. However, I recommend more sentences for autophagy itself at the molecular level, particularly of erythroid cells. Those description make readers to have more deep understanding of autophagy-targeted therapy.

3.     One of our questions is how we develop erythroid-lineage specific autophagy-targeted therapy. Please give your comments on this issue, which were partly answered by Dr. Kang et al. in 2023. I suggest to move sentences concerning GATA-1 from “Conclusion and future perspectives” to “Discussion” part.

Minor points:

1.     I suggest making the Table summarizing different approaches to reduce alpha-globin synthesis: siRNAs, miRNAs, CRISPR-Cas9 protocols, Rapamycin, etc. which the author described them in the text form and Figures. Then, this review would look more informative.

2.     Lines 38 – 39: Please give the definition of beta0 and beta+-Thalassemia, otherwise readers would not understand the following part of text.

3.     Line 102: Annotations Fig.3A, Fig.3B Fig.3C are not appropriate, since Figure 3 does not have panel A, B and C. Please follow the suggestion from the editing office.

4.     Lines 143 to 147: Something is wrong. Annotation (B) should be (C)?

5.     Line 247: “ice” must be “mice”.

Comments on the Quality of English Language

Only very minor revision is required.

Author Response

Reply to Reviewer #2

First of all, we like to thank the reviewer for her(his) work.

General comments. This manuscript describes the new approach for beta-Thalassemia using “Autophagy system” in erythroid cells as biomarkers or therapeutic targets. The aim and scope are important and precious for contributing to sustainable health care of patients with beta-Thalassemia. Structure of main part of the manuscript is logical and scientific enough. Many important previous reports are cited appropriately. Writing is excellent, easily understandable for scientists in broad fields. This review would give us many relevant knowledges and encouragement to contribute to improve patient’s life.

Answer to the general comments. We thank very much the reviewer for her(his) comments oping to have fairly addressed all of them.

Major points:

Point 1. Lines 181-238: Dysregulation of autophagy system in beta-Thalassemia is written only in abstract phrase. It is better to show data of up-regulation of autophagy in beta-Thalassemia accumulated in previous literatures.

Answer to point 1. This is a very good point. Thanks. We included the new chapter 3.3 (Autophagy and ineffective erythropoiesis in β-thalassemia) (pages 6, lines 201-234). The references 89-100 have been associated with this new chapter. 

Point 2. In general, there are few presentations for the molecular mechanism of autophagy. This is critical for understanding the cutting-edge of beta-Thalassemia treatment. We can see important progress in animal model and clinical trial in human in several parts of text. However, I recommend more sentences for autophagy itself at the molecular level, particularly of erythroid cells. Those description make readers to have more deep understanding of autophagy-targeted therapy.

Answer to point 2. Figure 4 has been implemented (page 5), including molecular players of autophagy steps. Figure 4 has been described in lines 153-178.

Point 3. One of our questions is how we develop erythroid-lineage specific autophagy-targeted therapy. Please give your comments on this issue, which were partly answered by Dr. Kang et al. in 2023. I suggest to move sentences concerning GATA-1 from “Conclusion and future perspectives” to “Discussion” part.

Answer to point 3. The sentences concerning GATA-1 have been moved from “Conclusion and future perspectives” to the “Discussion” part, as suggested. As far as the erythroid-lineage specific autophagy-targeted therapy, this issue is challenging and very important. We were unable to find “Kang et al., 2023). However, we briefly discuss issues related to therapy at page 12, lines 427-444.

Minor points:

Point 1. I suggest making the Table summarizing different approaches to reduce alpha-globin synthesis: siRNAs, miRNAs, CRISPR-Cas9 protocols, Rapamycin, etc. which the author described them in the text form and Figures. Then, this review would look more informative.

Answer to point 1. This is an excellent suggestion. Please find the Table 1 at page 10. We agree with the reviewer that this will help the reader.

Point 2. Lines 38 – 39: Please give the definition of beta0 and beta+-Thalassemia, otherwise readers would not understand the following part of text.

Answer to point 2. We have now defined beta0 and beta+ Thalassemia within the modified sentence  “In this case, particularly informative is the comparison of chromatograms using lysates from erythroid cells derived from …. shows representative chromatograms of erythroid cells of a beta0-thalassemia patient (Figure 2A) and a beta+-thalassemia patient (Figure 2B)” (page 2, lines 36-41).

Point 3. Line 102: Annotations Fig.3A, Fig.3B Fig.3C are not appropriate, since Figure 3 does not have panel A, B and C. Please follow the suggestion from the editing office.

Point 4. Lines 143 to 147: Something is wrong. Annotation (B) should be (C)?

Point 5. Line 247: “ice” must be “mice”.

Answer to point 3-5. All these issues have been considered and fixed.

In conclusion, we thank the reviewers for the hard work and for the suggestions, that were judged to be very useful for improving the scientific quality of our review paper. We hope that this manuscript will be now considered acceptable for publication in the Special Issue “Exclusive Review Papers in Autophagy-Second Edition” (Cells).

Waiting for your comments, we like to thank you in advance for your help.

Alessia Finotti and Roberto Gambari,

Department of Life Sciences and Biotechnology

University of Ferrara

Ferrara, May 9, 2024

Reviewer 3 Report

Comments and Suggestions for Authors

In this nicely written review the authors summarize recent evidence on the role of autophagy via the Ulk1 pathway in order to control accumulation of alpha globin chains and subsequent toxicity. The authors also highlight potential avenues for treatment of beta- thalassemia by modulating Ulk1 expression f.e. via modulation of the mTOR pathway. This review provides an up to date summary of the current knowledge regarding this autophagy in thalassemia major and nice builds the bridge from in vitro observations to mouse models and data in humans along with a discussion of ongoing trials in that specific field.

Minor:   Toxicity and ineffective erythropoiesis are also caused by excessive iron absorption and tissue iron accumulation which causes oxidative stress and lipid peroxidation via the Fenton reaction. This should be stated in detail. It would be also of interest to discuss how iron chelation therapy may impact on mTOR activity given that activity of HIF-alpha (as a consequence of treatment with desferrioxamine) may also affect mTOR expression.

Please explain by one or two additional sentence how the increase in hemoglobin F will ameliorate toxicity by alpha chain accumulation.

It would be also if interest to know if there are other diseases where Ulk1/autophagy mediated  pathways are involved or beneficially modified by pharmacological interventions.

Line 325:  This sentence appears to be truncated.

Comments on the Quality of English Language

good

Author Response

Reviewer #3

General comments. In this nicely written review, the authors summarize recent evidence on the role of autophagy via the Ulk1 pathway in order to control accumulation of alpha globin chains and subsequent toxicity. The authors also highlight potential avenues for treatment of beta-thalassemia by modulating Ulk1 expression i.e. via modulation of the mTOR pathway. This review provides an up to date summary of the current knowledge regarding this autophagy in thalassemia major and nice builds the bridge from in vitro observations to mouse models and data in humans along with a discussion of ongoing trials in that specific field.

Answer to the general comments. We thank very much the reviewer for her(his) comments oping to have fairly addressed all of them.

Minor points.

Point 1. Toxicity and ineffective erythropoiesis are also caused by excessive iron absorption and tissue iron accumulation which causes oxidative stress and lipid peroxidation via the Fenton reaction. This should be stated in detail. It would be also of interest to discuss how iron chelation therapy may impact on mTOR activity given that activity of HIF-alpha (as a consequence of treatment with desferrioxamine) may also affect mTOR expression.

Answer to point 1. This is a very important point. To discuss issues related to iron and ineffective erythropoiesis and autophagy, the following sentence has been added: “Finally, in the case of transfusion-dependent beta-thalassemia, the role of iron in ineffective erythropoiesis and autophagy should be considered [96] …..with respect to possible impact of iron chelators on the autophagic process considering the use of these agents for the treatment of iron overload diseases, such as beta-thalassemia [100]” (page 6, lines 222-234).   

Point 2. Please explain by one or two additional sentence how the increase in hemoglobin F will ameliorate toxicity by alpha chain accumulation.

Answer to point 2. In order to follow this suggestion, this sentence has been added: “Reduction of the excess of free α-globin chains in b-thalassemia cells can be also obtained following g-globin gene activation and fetal hemoglobin (HbF) induction ….. In this case, increased g-globin gene expression leads to increased production of g-globin. This contributes to decreasing free a-globin by the formation of the a2g2 tetramers (HbF)” (page 3, lines 113-117).   

Point 3. It would be also if interest to know if there are other diseases where Ulk1/autophagy mediated pathways are involved or beneficially modified by pharmacological interventions.

Answer to point 3. This is a very important point. To follow the suggestion, we include chapter 3.2 (Autophagy in human diseases) and the sentence “In cancer, autophagy has been suggested to behave as tumor suppressor as well tumor promotion ….. ……. Other examples of diseases associated with autophagy are several metabolic diseases, Crohn's disease, Lysosomal storage disorders (LSDs)[88]” (pages 5 and 6, lines 180-199. The new references 66-88 have been added to sustain the text.

Point 4. Line 325:  This sentence appears to be truncated.

Answer to point 4. Right. Tanks. Found and fixed.

In conclusion, we thank the reviewers for the hard work and for the suggestions, that were judged to be very useful for improving the scientific quality of our review paper. We hope that this manuscript will be now considered acceptable for publication in the Special Issue “Exclusive Review Papers in Autophagy-Second Edition” (Cells).

Waiting for your comments, we like to thank you in advance for your help.

Alessia Finotti and Roberto Gambari,

Department of Life Sciences and Biotechnology

University of Ferrara

Ferrara, May 9, 2024

Round 2

Reviewer 1 Report

Comments and Suggestions for Authors

Authors have clarified the concerns